# Different Infectivity of Mediterranean and Southern Asian Tomato Leaf Curl New Delhi Virus Isolates in Cucurbit Crops

**DOI:** 10.3390/plants11050704

**Published:** 2022-03-06

**Authors:** Thuy T. B. Vo, Aamir Lal, Phuong T. Ho, Elisa Troiano, Giuseppe Parrella, Eui-Joon Kil, Sukchan Lee

**Affiliations:** 1Department of Integrative Biotechnology, Sungkyunkwan University, Suwon 16419, Korea; bichthuy251188@gmail.com (T.T.B.V.); aamirchaudhary43@gmail.com (A.L.); hophuongk59sinhhoc@gmail.com (P.T.H.); 2Institute for Sustainable Plant Protection of the National Research Council (IPSP-CNR), 80055 Portici, Italy; elisa_troiano@libero.it; 3Department of Plant Medicals, Andong National University, Andong 36729, Korea

**Keywords:** tomato leaf curl New Delhi virus, begomovirus, cucurbit crops

## Abstract

Tomato leaf curl New Delhi virus (ToLCNDV) became an alerting virus in Europe from 2017 to 2020 because of its significant damage to *Cucurbitaceae* cultivation. Until now, just some cucurbit crops including sponge gourd, melon, pumpkin, and cucumber were reported to be resistant to ToLCNDV, but no commercial cultivars are available. In this study, a new isolate of ToLCNDV was identified in Pakistan and analyzed together with ToLCNDV-ES which was previously isolated in Italy. Furthermore, infectious clones of two ToLCNDV isolates were constructed and agroinoculated into different cucurbit crops to verify their infectivity. Results showed that both isolates exhibited severe infection on all tested cucurbit (>70%) except watermelon. Thus, those cultivars may be good candidates in the first step of screening genetic resources for resistance on both Southeast Asian and Mediterranean ToLCNDV isolates. Additional, comparison pathogenicity of different geographical ToLCNDV isolates will be aided to understand viral characterization as such knowledge could facilitate breeding resistance to this virus.

## 1. Introduction

Tomato leaf curl New Delhi virus (ToLCNDV) is a bipartite member of the genus *Begomovirus* belonging to *Geminiviridae* family. The virus comprises two circular single-strand DNA segments designated as DNA A and DNA B. The DNA A component includes six ORFs, encodes all information for viral encapsidation, replication, and produces virions. DNA B contains two ORFs and is required for symptom expression, systemic movement, nuclear localization, and systemic infection. ToLCNDV was first identified in tomato from India around 25 years ago and subsequently distributed across the Indian subcontinent, East and Southeast Asia [1,2,3,4]. This virus only appeared in Asian countries until it was first reported in Mediterranean countries in 2012 then rapidly spread through Europe [5,6,7,8,9,10,11]. This virus has broad host ranges and members of the *Solanaceae* and *Cucurbitaceae* families are mainly known as this virus’s natural host. ToLCNDV-infected plants show typical symptoms including severe curling, yellow mosaic and vein swelling of young leaves, shortening of internodes, rough skin, and reduced size of fruit [12,13]. Since the first report, ToLCNDV has caused serious problems affecting tomato production in the India subcontinent and cucurbit cultivation in the Mediterranean countries. This virus has been included as an alerting virus in the European and Mediterranean Plant Protection Organization Plant alert list from 2017 to 2020. Besides the natural transmission by whitefly (*Bemisia tabaci*) in a persistent manner, this virus was also reported to transmit through seeds or mechanical sap-inoculation [14,15]. With the possibility of alternative types of transmission, many important crops production could suffer massive losses due to ToLCNDV infection.

The GenBank database reports over 500 sequences of ToLCNDV isolates found in Asia and the Mediterranean Basin. The Mediterranean isolates of this virus have been categorized as a different strain and have different infection abilities in important crops such as tomato compared with the Asian isolates [16,17].

Application of insecticide and chemicals for controlling insect vector populations has been a leading approach to control begomoviruses. However, unregulated use of insecticides leads to insect resistance to overcome any treatment, thus crop cultivation becomes difficult [18]. The insect controlling approach also presented a limitation to managing ToLCNDV pathogens and needs another effective alternative. Recently, screening for resistance cultivars has become the main approach in attempting to manage ToLCNDV pathogens. Until now, some studies have reported ToLCNDV resistance in the sponge gourd (*Luffa aegyptiaca*), melon (*C. melo*), pumpkin (*C. moschata*) [19,20,21], and cucumber (*Cucumis sativus*), recently [22]. However, there is a need to study and screen more resistance cultivars against ToLCNDV due to the wide broad host range of this virus as well as less advanced research in the resistance genotype. In this research, two different ToLCNDV isolated from Italy (ToLCNDV-ES) and Pakistan (ToLCNDV-India) were studied to compare their pathogenicity among cucurbit crops. Infectious clones of both isolates were constructed and agroinoculated in pumpkin (*Cucurbita moschata*), zucchini (summer squash, *Cucurbita pepo*), melon, and watermelon (*Citrulluslanatus*). Considering the differences in pathogenicity with ToLCNDV isolates from different geographical areas can offer a good candidate for genotype resistance cultivars and provide deeper knowledge about the characterization of this virus.

## 2. Results

### 2.1. Viral Isolation and Phylogenetic Analysis of Two ToLCNDV Isolates

In 2019, yellowing *V. rosea* leaves were observed and collected in Pakistan during a survey for begomovirus detection analysis. Total DNA was purified and used for PCR amplification using Beg-F/Beg-R (data not shown). The 1 kb PCR products were cloned and sequenced. BLAST analyses showed that the detected virus sequence shared 95% nucleotide similarity with ToLCNDV isolates from sponge gourd in Pakistan (KY933709). Through RCA and sequencing, the full-length genome of ToLCNDV isolates from Pakistan (named ToLCNDV-India) was obtained and submitted to GenBank. The complete genomes of ToLCNDV-India DNA A (Accession number: MT085663) and DNA B (Accession number: MT085664) comprise of 2739 and 2691 nucleotides, respectively. The number and arrangement of the ORFs and intergenic region were identical to those of the ToLCNDV isolates in tomato (EF068246). The target amplicon from PCR reaction on Italy infected pumpkin samples verified a previously reported viral sequence (Accession number: MF688670) [10] and can be used for further studies.

A total of 11 ToLCNDV full length sequence isolated from Europe/Mediterranean and 38 sequences isolated from Asia on different hosts including two sequences generated from Italy and Pakistan in this study were aligned using MEGA 7 software and constructed phylogenetic trees through the neighbor-joining method (Figure 1). Sequences of both isolates in our study formed a different clade on a phylogenetic tree. ToLCNDV-India DNA A is closely related (98%) with ToLCNDV isolated from tomato in India, while DNA B showed high similarity (99%) with ToLCNDV isolated from chayote in India and tomato in Bangladesh. Furthermore, sequences of both ToLCNDV-ES DNA A and DNA B components are 100% similar to ToLCNDV isolated from chili in Italy and share the same clade with other ToLCNDV isolates from Mediterranean countries. These results indicated that ToLCNDV isolated from Pakistan in our study is a variant of the ToLCNDV India strain, also genetically distant to ToLCNDV identified from Italy—a new isolate of ToLCNDV-ES strain.

### 2.2. Infectivity of Two ToLCNDV Isolates’ Infectious Clone

An infectious clone of two ToLCNDV isolates was constructed using the tandem repeat fragment method. Based on the general genome structure of DNA A and DNA B segments, new primer sets were designed to generate ~1.1 mer (DNA A) and ~1.2 mer (DNA B) construction and ligated with pCAMBIA 1303 vector (Figure 2). The recombinant plasmid was transformed into *A. tumefaciens* GV3101 and agroinoculated into *Nicotiana benthamiana* to confirm their infectivity.

All *N. benthamiana* plants inoculated with ToLCNDV-ES and ToLCNDV-India clone induced typical phenotypes compared with mock plants at 21 dpi (Figure 3A). Mock plants inoculated with pCAMBIA 1303 empty induced no virus symptoms. PCR and Southern blot results showed that both ToLCNDV-ES and ToLCNDV-India infected and replicated in *N. benthamiana* (Figure 3B,C). These results indicated that both constructed clones of ToLCNDV isolates were infectious and can be used for further experiments.

Then, ToLCNDV-ES and ToLCNDV-India were agroinoculated to four cucurbit crops which include pumpkin, zucchini, melon, and watermelon to check their infectivity. Each species was grown using commercial seeds from Italy and Korea. Three weeks after inoculation, except watermelon, three other crops tested positive for ToLCNDV-ES and ToLCNDV-India infection (Figure 4A and Table 1). All positive plants induced viral symptoms. Both ToLCNDV clones inoculated pumpkin and melon leaves induced yellow mosaic and curling symptoms. For zucchini plants, ToLCNDV-India infected plants showed heavier curling and mosaic symptoms than those of ToLCNDV-ES infected plants, demonstrating that ToLCNDV-ES had lower pathogenicity than ToLCNDV-India in this crop. Only watermelon presented no ToLCNDV infection including Italy or Korean cultivars. The infectivity of ToLCNDV was confirmed by PCR and Southern blot (Figure 4B,C). The specific amplicon was detected on PCR products of pumpkin, zucchini, and melon. Additionally, similar results of Southern blot were achieved on inoculated plants. The results indicated that both ToLCNDV isolates can infect cucurbit crops except watermelon.

### 2.3. Inoculation of Two ToLCNDV Isolates on Different Watermelon Cultivars

A screening study was conducted using different watermelon cultivars to confirm whether ToLCNDV truly has no pathogenicity on watermelon. Among eight cultivars used in this study, none of them showed ToLCNDV symptoms (Figure 5 and Table 2). All tested plants presented a normal phenotype at 30 dpi. According to PCR results, only ToLCNDV infected *N. benthamiana* plants used as positive plants were positive while others were negative, regardless of whether they were Korean or Italian cultivars. These results demonstrated that both ToLCNDV-ES and ToLCNDV-India did not infect watermelon cultivars used in this study. Therefore, those watermelon cultivars could be useful for ToLCNDV resistance in cucurbit crops.

## 3. Discussion

Begomovirus, the largest genus in the *Geminiviridae* family, has been identified to cause significant problems for vegetable and legume crop production in many regions [23]. Among the begomoviruses reported across the world, ToLCNDV has high economic importance in horticultural production. This virus was first identified in India in 1995 and quickly spread across the Indian subcontinent, Southeast Asia, and East Asia, causing tremendous trouble in production of many crops. In the Mediterranean area, ToLCNDV has been recognized as an emerging threat in the production of cucurbit crops since their first report in 2012 in Spain [5,6,7,8,9,10,11]. More than 20% of zucchini and melon in Spain [21,24], and 80% of pumpkin in Italy were damaged by a ToLCNDV outbreak [10,25]. Categorized as a different strain than Asian ToLCNDV, the Mediterranean isolates showed different pathogenicity on some plants such as tomato. ToLCNDV-ES isolated from Spain has lower incidence on tomato which is consistent with the reduced transmission efficiency in this plant. A recent report showed that transmission efficiency was significantly higher in zucchini (96%) compared to tomato (2%) by whitefly [26], so tomato crop can be a potential resource to against ToLCNDV. However, host range extension of ToLCNDV aided by its evolution and the possibility of a different transmission mode are making this virus become a significant threat to cultivations of important crops [27]. Therefore, searching for ToLCNDV-resistant resources in other species plays an essential role in overcoming this virus impact.

In our research, two ToLCNDV isolates from Italy and Pakistan named ToLCNDV-ES and ToLCNDV-India were used to study the host range and compare pathogenicity in order to understand their characterization as well as initiate screening for resistance sources of this virus. Phylogenetic analysis showed genetic distance from Europe/Mediterranean and Asia together with other isolates. Infection clones were constructed for both isolates after recombination plasmids were transformed into *Agrobacteria*. Agroinoculation on *N. benthamiana* achieved an almost 100% infection rate. Leaf curling, mosaic yellowing, and stunting symptoms that were observed indicated that both clones of two isolates presented a similar pathogen pattern with natural virus. Then, infection results on four cucurbit crops demonstrated that these clones represented a serious pathogenicity to pumpkin, zucchini, and melon as shown previously for other Mediterranean or Asian isolates. Moreover, ToLCNDV-India induced more adverse symptoms compared to those of ToLCNDV-ES in zucchini plants, indicating that ToLCNDV-India had higher pathogenicity in this species. These results showed a similarity with previous publications in zucchini plants in spite of genetic differences among each isolate [17,28].

Both isolate clones exhibited no adaption to infect watermelon, suggesting that this species is not a susceptible host for ToLCNDV. However, Fortes reported an almost 80% watermelon infection with ToLCNDV-[ES-Alm-661-Sq-13] isolated from Spain in 2016. A recent publication revealed that natural ToLCNDV infected field watermelon with yellowing and downward curling symptoms in India [29]. This indicated that watermelon still is a susceptible host for ToLCNDV induced disease. The cultivars used in this study might contain genetic variability of the crop species required for the resistant genotype. Moreover, with a large number of cultivars in more than 96 countries, each cultivar could provide different properties for virus infection. Finding ToLCNDV resistance in watermelon will not only prevent economic damage of watermelon production but also provide a good option against different viral infections on other important cucurbit production, as inbred cucumber did with papaya ring spot virus and watermelon mosaic virus resistance [30,31]. Previous studies reported ToLCNDV resistance in sponge gourd, melon, and pumpkin [19,32] and recently in cucumber [22]. Our study recommends preliminary candidates to screen more resistance cultivars in the very first step to prevent ToLCNDV. Further studies with genomic analysis using these materials can facilitate a breeding program and propose commercial cultivars resistant or tolerant to ToLCNDV infection. Additionally, a comparison between the pathogenicity of two ToLCNDV isolates from different geographical regions can clarify which viral characteristics are responsible for this pathogenicity in the future.

## 4. Materials and Methods

### 4.1. Virus Sources

Leaf samples from *Vinca rosea* showing yellowing symptoms were collected in Pakistan during surveys in 2019, then viral DNA was extracted using Viral Gene-spin™ DNA/RNA Extraction kits (iNtRON Biotechnology, Inc., Seongnam, Korea). The total obtained DNA was used for PCR amplification using primer sets as previously published [33,34]. The PCR products were sequenced using the commercial service Macrogen (Seoul, Korea). Rolling circle amplification (RCA) was conducted using the TempliPhi™ 100 Amplification kits (Cytiva, Marlborough, MA, USA) following the manufacturer’s instructions to obtain a full-length genome sequence for ToLCNDV-India. PCR amplification was conducted based on the RCA template and newly designed full-length primers to obtain a full sequence of ToLCNDV-India. The target amplicon was cloned into the pGEM T-easy vector (Promega, Madison, WI, USA) and sequenced by Macrogen. A recombinant plasmid was sequenced and submitted to the GenBank database. ToLCNDV-ES infected pumpkin samples detected in Italy [10] were used as a template for infectious clone construction.

### 4.2. Phylogenetic Analysis

Phylogenetic analysis was conducted using the full-length DNA A and DNA B of ToLCNDV containing sequences obtained in our studies (ToLCNDV-ES from Italy and ToLCNDV-India from Pakistan). In total, 50 full-length sequences for each DNA component identified on different hosts from Asia and Mediterranean countries were derived from GenBank and used to indicate the relationship among different ToLCNDV isolates. By applying the ClustalW algorithm, sequences were aligned using MEGA 7 software and a phylogenetic tree was generated using the neighbor-joining methods [35]. The tree branches were bootstrapped with 1000 replications.

### 4.3. ToLCNDV Infectious Clone Construction

Infectious clones of two ToLCNDV isolates were obtained by constructing a partial tandem repeat of the full-length viral DNA as previously described [36].

Specific primer sets were designed to detect two fragments of each DNA component through PCR amplification (Appendix A). Then, each partial fragment of ToLCNDV DNA A and DNA B was cloned into the pGEM T-easy vector and sequenced for confirmation. After digestion by restriction enzyme, both fragments of each DNA were ligated to pCAMBIA 1303 vector to generate 1.1-mer for DNA-A and 1.2-mer for DNA B constructs. The infectious constructs of ToLCNDV DNA A and DNA B were passed on *Agrobacterium tumefaciens* strain GV3101 using the freeze–thaw transformation method.

### 4.4. Inoculation Tests

*Nicotiana benthamiana* and two different cultivars of each species including pumpkin, zucchini, melon, and watermelon were grown until three weeks with fully expanded true leaves for inoculation. Cell cultures of each clone containing DNA-A and DNA-B components were grown at 28 degrees with LB broth with rifampicin, gentamycin, and kanamycin until OD at 600 nm reached 1.0 and then inoculated on the apical side of plants using plastic dropping pipette and needles. All inoculated plants were kept in a plant growth room with a photoperiod of 16 h light and a target air temperature set at 28/22 °C day/night and the symptoms were observed after three weeks post inoculation. The ToLCNDV infection was verified by conventional PCR and symptomatic phenotype inspection in experimental plants.

Eight commercial seeds of watermelon from Italy (Nevada-F1, Brera-F1, Talete-F1, Crimson Sweet cv.) and Korea (Black honey, Honey gold, Sinseolgang 102, Plus Honey cv.) were used for checking ToLCNDV seed transmission.

### 4.5. Viral DNA Detection

All ToLCNDV inoculated plants were analyzed by PCR amplification with specific primer sets I-A-1/I-B-2 for ToLCNDV-ES and Pa-A-1/Pa-B-1 for ToLCNDV-India (Appendix A). After three weeks post inoculation, new leaves were taken and viral DNA was extracted using Viral Gene-spin^TM^ Viral DNA/RNA Extraction Kit following the manufacturer’s instructions. For PCR reaction, the mixture was prepared with the following reagents: 10 µL of 1 × *AccuPower*^®^ PCR Mastermix (Bioneer), 1 µL of template DNA, 1 µL of primer forward (10 pM), 1 µL of primer reverse (10 pM), and distilled water to reach a total volume of 20 µL. The conditions for PCR reaction were 95 °C for 3 min, 34 cycles (denaturation at 95 °C for 30 s, annealing at 58 °C (both isolates) for 30 s and extension 72 °C for 1 min (DNA A) and 2 min (DNA B) followed with final extension at 72 °C for 5 min. The PCR products were electrophoresed on 1% agarose gel and stained with ethidium bromide for the amplified target size.

### 4.6. Southern Hybridization Blot

Southern hybridization was conducted to confirm the viral replication of ToLCNDV in experimental plants using leaf samples after 21 days post inoculation. DNA was extracted by Sodium-Tris-EDTA (STE) buffer [37] and used as an object for Southern blotting [38]. Total DNA isolated from plant tissues was electrophoresed on 1% agarose gels then transferred to nylon membranes (Amersham Hybond-N+ membrane, Cytiva) by capillary transfer for up to 16 h. The amplified DNA fragments were gel purified and hybridized with a [α^32^P]-dCTP using the Rediprime II Random Primer Labeling System (Cytiva) at 65 °C for 16 h. The membrane was exposed to X-ray film for 24 h in a −80 °C freezer after washing.

## Figures and Tables

**Figure 1 plants-11-00704-f001:**
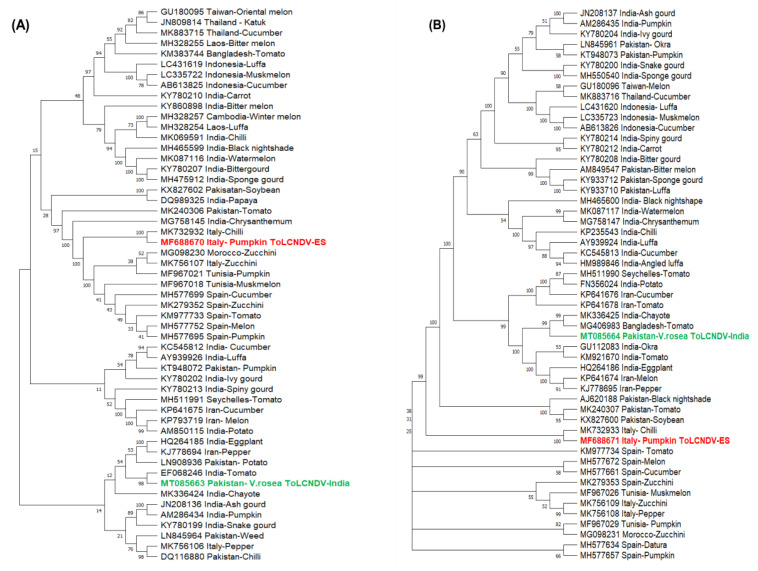
Phylogenetic analysis of two ToLCNDV isolates from Italy (ToLCNDV-ES) and Pakistan (ToLCNDV-India). Phylogenetic relationships were generated using the neighbor-joining methods in the MEGA 7 program. Numbers at branch internodes represent bootstrap values (1000 replicates). The ToLCNDV isolates from Italy (Accession number MF688670/MF688671) and Pakistan (Accession number: MT085663/MT085664) used in this study are highlighted by red and green colors, respectively. (**A**) DNA A component. (**B**) DNA B component.

**Figure 2 plants-11-00704-f002:**
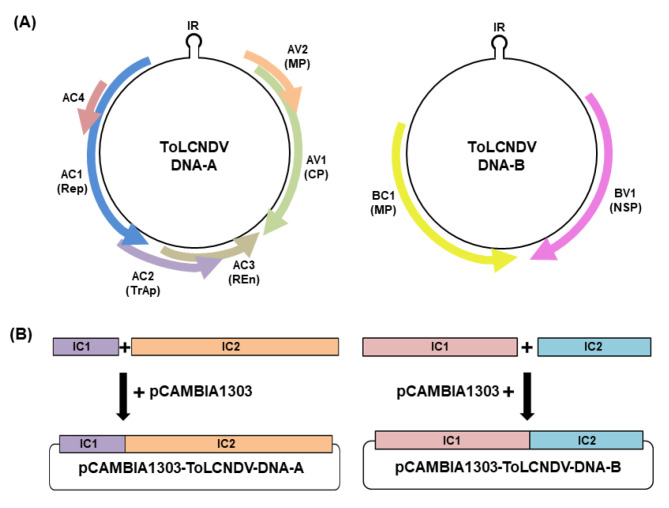
Genome structure and infectious clone construction of ToLCNDV. (**A**) Genome structure of DNA A and DNA B of ToLCNDV. (**B**) Scheme for ToLCNDV infectious clone construction by partial tandem repeat construction method. Two fragments of each component were generated and ligate with pCAMBIA 1303 vector to harbor a recombinant plasmid.

**Figure 3 plants-11-00704-f003:**
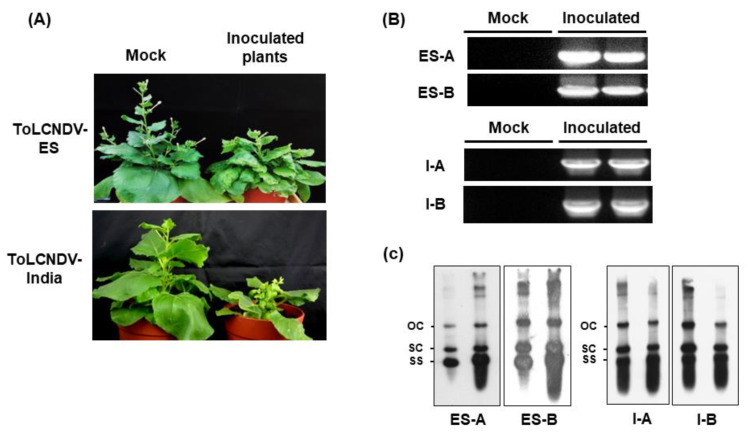
Infectivity of two ToLCNDV isolates in *Nicotiana benthamiana* at 21 dpi. (**A**) Phenotypes of infected *N. benthamiana* with ToLCNDV-ES and ToLCNDV-India. The infected plants showed severe stunning, leaf curling symptoms. (**B**) Detection results of inoculated plants by PCR. (**C**) Southern hybridization blotting results. PCR bands are used as probe. ES-A, ES-B, I-A, and I-B are abbreviations of amplicons that were detected from ToLCNDV-ES DNA A, DNA B and ToLCNDV-India DNA A, DNA B, respectively. OC, open circular dsDNA; SC, supercoiled dsDNA, SS, ssDNA.

**Figure 4 plants-11-00704-f004:**
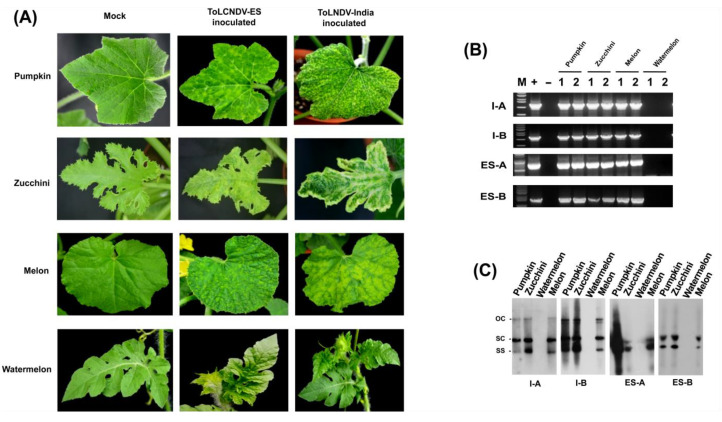
Infectivity of two ToLCNDV isolates in cucurbit plants after three weeks post inoculation. (**A**) Phenotypes of inoculated plants on different cultivars of pumpkin, zucchini, melon, and watermelon. Infected plants of all test species except watermelon showed curling and yellow mosaic symptoms. (**B**) ToLCNDV detection results by PCR. Lane M: ladder; Lane +: positive control; Lane −: negative control. (1) Korean cultivar, (2) Italian cultivar. (**C**) Southern blot results.

**Figure 5 plants-11-00704-f005:**
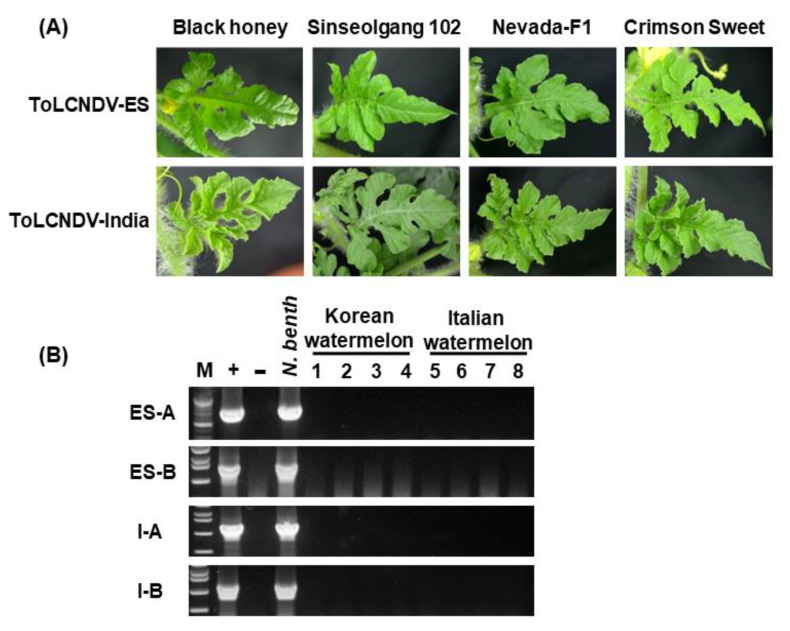
Different watermelon cultivars inoculated with two ToLCNDV isolates via agroinoculation. (**A**) Normal phenotype of inoculated plants. Blackhoney and Sinseogang102 as Korean watermelon, Nevada-F1 and Crimson Sweet as Italian watermelon. (**B**) PCR results using specific primers for each DNA component of two ToLCNDV isolates. *N. benthamiana* was used to test positive with ToLCNDV. Lane M: ladder; Lane +: positive control; Lane −: negative control.

**Table 1 plants-11-00704-t001:** Infectivity of two ToLCNDV isolates on *Cucurbitaceae* plants.

Isolates	Plant	Infectivity *	PCR	Symptom
ToLCNDV-India	Pumpkin (cv. Shin Asia Matdol)	7/7	+	Curling, yellow mosaic
	Pumpkin (cv. Napoletana)	7/7	+	Curling, yellow mosaic
	Zucchini (cv. Saeroun)	6/7	+	Severe curling, yellow mosaic
	Zucchini (cv. Romanesco)	5/7	+	Severe curling, yellow mosaic
	Melon (cv. Earl’s Mountain PMR)	6/7	+	Curling, yellow mosaic
	Melon (cv. RetatoDegliOrtolani)	5/7	+	Curling, yellow mosaic
	Watermelon (cv. Yeoreumen)	0/9	-	--
	Watermelon (cv. Sugar Baby)	0/9	-	--
ToLCNDV-ES	Pumpkin (cv. Shin Asia Matdol)	7/7	+	Mild curling, yellow mosaic
	Pumpkin (cv. Napoletana)	6/7	+	Mild curling, yellow mosaic
	Zucchini (cv. Saeroun)	5/7	+	Curling, yellow mosaic
	Zucchini (cv. Romanesco)	5/7	+	Curling, yellow mosaic
	Melon (cv. Earl’s Mountain PMR)	5/7	+	Mild curling, yellow mosaic
	Melon (cv Retato Degli Ortolani)	7/7	+	Mild curling, yellow mosaic
	Watermelon (cv. Yeoreumen)	0/10	-	--
	Watermelon (cv. Sugar Baby)	0/10	-	--

* Number of infected plants/number of inoculated plants. Infectivity and symptoms were observed at 21 dpi for all tested plants.

**Table 2 plants-11-00704-t002:** Infectivity of two ToLCNDV isolates on different watermelon cultivars.

Cultivars	Infectivity *	
	ToLCNDV-India	ToLCNDV-ES
Black honey	0/7	0/7
Honey gold	0/7	0/7
Sinseolgang102	0/7	0/7
Plus Honey	0/7	0/7
Nevada-F1	0/7	0/7
Brera-F1	0/7	0/7
Talete-F1	0/7	0/7
Crimson Sweet	0/7	0/7

* number of infected plants/number of inoculated plants. Infectivity and symptoms were observed at 21 dpi for all tested plants.

## Data Availability

Data is contained within the article or Appendix A.

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
