# Peer review of "Different Infectivity of Mediterranean and Southern Asian Tomato Leaf Curl New Delhi Virus Isolates in Cucurbit Crops"

_plants, 2022, doi:10.3390/plants11050704_

Round 1

Reviewer 1 Report

From a first reading, in my opinion the article titled “Different infectivity of tomato leaf curl New Delhi virus the Mediterranean and Southern Asian isolates in cucurbit crops” - Manuscript ID: plants-1607975, by Thuy Thi Bich Vo et al., submitted to section: Plant Molecular Biology is interesting because ToLCNDV actually represent a threat for horticultural production worldwide; so, it is very important to develop a suitable crop management, applying genetic resistance strategies.

I think the presented manuscript have a good potential and it is adequate for publishing in Plants, but some minor corrections are need:

Title: In my modest opinion it would be appropriate to change the title as follows: Different infectivity of Mediterranean and Southern Asian tomato leaf curl New Delhi virus isolates in cucurbit crops

Line 15: please add “,” after ToLCNDV and change “were” with “was”.

Line 16: please change “analyze” with “analyzed” and “were” with “was”.

Line 17: please change “isolates ToLCNDV” with “ToLCNDV isolates”

Line 17: please change “check” with “verify”

Line 19: please add “isolates” at the end of sentence.

Line 20: please change “isolates ToLCNDV” with “ToLCNDV isolates”

Lines 26-28: please add more informations on ToLCNDV genome, as follow: DNA-A encodes all information for viral encapsidation and replication, produces virions, and can replicate autonomously (see Rogers SG, Bisaro DM, Horsch RB et al., 1986. Tomato golden mosaic virus A component DNA replicates autonomously in transgenic plants. Cell 45, 593–600). DNA-B cannot replicate in the absence of DNA-A, but it is required for symptom expression, systemic movement, nuclear localization and systemic infection (see Sanderfoot AA, Lazarowitz SG, 1996. Getting it together in plant virus movement: cooperative interactions between bipartite geminivirus movement proteins. Trends in Cell Biology 6, 353–8.).

Line 28: please write “ORFs”

Line 33: please change “display” with “show”

Line 35: add a space after “[12,13]”

Line 35: please change “big” with “serious”

Line 39: please change “persistently” with “in persistent manner”

Line 42: please change the sentence “ToLCNDV isolates from Asia and the Mediterranean region have over 500 sequences in the GenBank database” with “In GenBank database are reported over 500 sequences of ToLCNDV isolates found in Asia and the Mediterranean Basin.”

Line 47: please rewrite the sentence more clearly

Line 54: please delete “(Cucumis melo)” and add a space in “Citrullus lanatus”.

Line 61: change “isolated” with “purified”

Line 62: change “searches” with “analyses”

Lines 64-68: please rewrite more clearly the sentence.

Lines 69-70: delete this sentence “Phylogenetic analysis was conducted using full-length DNA A and DNA B sequences of various ToLCNDV including sequences obtained in this study” and specify the number of DNA A and DNA B obtained in this study were used for phylogenetic analysis

Line 72: change “had closest” with “is closely related with”

Line 73: change “to” with “with”

Lines 74-76: the sentence is unclear, please rewrite better

Line 78: please change “isolates ToLCNDV” with “ToLCNDV isolates”

Line 78: please change “isolated” with “isolates”

Line 80-81: please change “isolates ToLCNDV” with “ToLCNDV isolates”

Line 105: delete “None of the inoculated plants showed 105 disease symptoms”, because it is redundant

Line 106: change “band” with “amplicon”

Line 110: please change “isolates ToLCNDV” with “ToLCNDV isolates”

Line 111: please change “included” with “of”

Line 116: please change “isolates ToLCNDV” with “ToLCNDV isolates”

Line 118: change “infection of ToLCNDV” with “ToLCNDV symptoms”

Line 120: change “showed” with “resulted”

Line 122: change “initiative option in the search” with “useful”

Line 124: add “cultivars” after watermelon and change “infection of ToLCNDV” with “ToLCNDV symptoms”

Line 126 and 129: change “isolates ToLCNDV” with “ToLCNDV isolates”

Line 137: add the following reference - Panno, S., A. G. Caruso, E. Troiano, M. Luigi, A. Manglli, T. Vatrano, G. Iacono et al. "Emergence of tomato leaf curl New Delhi virus in Italy: estimation of incidence and genetic diversity." Plant Pathology 68, no. 3 (2019): 601-608.

Line 138-139: change “alternative types of transmission” with “different transmission modes” and rewrite the sentence better

Line 140: change “isolated” with “isolates”

Line 141: add “, in order” after “pathogenicity”

Line 146: “good infection” ??? please clarify!

Line 154: delete “cultivated”

Line 161: change “isolates ToLCNDV” with “ToLCNDV isolates”

Line 166: please change “The isolated DNA” with “The total DNA obtained”

Line 168: add a space before “Rolling”

Line 173: please delete “supported by Dr Parrella in Italy”

Line 176: please add the reference for ClustalW program

Line 178: please add an empty line before the paragraph “4.3. ToLCNDV infectious clone construction”

Line 189: please add space in “600nm”

Line 191: please change the sentence “16/8 h light/dark periods, 22-28 degrees” with “with a photoperiod of 16 h light and a target air temperature set at 28/22 °C day/night”. And then modify the sentence “for three weeks to observe symptoms” with “; the symptoms were observed after three weeks post inoculation”.

Lines 191-192: please change the sentence “The consequence of ToLCNDV 191 infection was analyzed using PCR” with “The ToLCNDV infection was confirmed by PCR”

Line 194: please change “infectivity experiment” with “seed transmission”

Line 196: please change “inoculation was confirmed” with “inoculated plants were analyzed” and delete “;” after “ToLCNDV-ES” and add “and”

Line 197:  change “of” with “post”

Line 200: add “reach” after “to”

Line 202: change “and terminated with” with “following a”

Lines 205-206: please rewrite better the sentence “Leaves from plants 21 days post-inoculation were harvested for DNA extraction and analyzed using Southern hybridization to confirm the viral replication of ToLCNDV in experimental plants.”

Finally, although the language used in the manuscript is easy to follow and understandable, however, the manuscript should be carefully revised for grammar and English use, since some mistakes were found throughout the whole paper.

Reviewer 2 Report

The manuscript submitted by Thuy T. B. Vo and coworkers, entitled "Different infectivity of tomato leaf curl New Delhi virus the Mediterranean and Southern Asian isolates in cucurbit crops" contains results of infectivity tests performed on different cucurbit crops with two different ToLCNDV isolates (from Pakistan and Italy). Infection of host plants in the trials was verified by PCR and Southern blot. In addition, the isolates were compared at the molecular level with other isolates available in GenBank. Infection with the virus was successful in pumpkin, zucchini, and melon, while infectivity tests in eight different watermelon cultivars resulted in no infection. Based on the results obtained in watermelon, possible sources of resistance/tolerance are discussed. However, some points need to be further elaborated.

General comments:

The M&M section is too short and does not contain enough information, which is then presented in the Results section. An example is the phylogenetic analysis where most of the information is presented under the Results section (which isolates were used) but not under M&M. The descriptions of the figures are too brief and cannot be explained by themselves without reading the data in the main text. In addition, many of the abbreviations used in the figures are not explained, including the lack of any description/explanation in the supplementary material.

Specific comments:

Abstract – line 14 – Cucurbitaceae – should be in italic

  • Line 21 – resistance on this virus

Introduction – lines 27 and 28 – ORFs

  • Line 32-33 with members of the Solanace and Cucurbitaceae family as the virus main hosts.
  • Line 35 - . Since; delete big

Recently Janssen et al. “Host species-dependent transmission of tomato leaf curly New Delhi virus-ES” was published in Plants. It would be useful to include the results of this work and incorporate them in the introduction, but also in the discussion section.

Line 52 – naming the isolate ToLCNDV-ES is a little bit confusing, since addition of ES resemble to the isolate(S) originating from Spain. Please reconsider to change name of the Italian isolate.

Line 54 – Cucumis melo, Citrullus lanatus

Results

Line 59 – of two ToLCNDV isolates

Line 66 – isolates from tomato (please provide GenBank acc. no); please give detailed description how phylogenetic tree was constructed using which isolates for comparison under the M&M section

Figure – full description is missing – explanation that each virus isolate used in comparison is represented by isolate name and corresponding GenBank acc. no.

Line 80 – of two ToLCNDV isolates

Line 90 – symptoms. PCR and

Figure 3 – description of abbreviations used is (I-A, I-B etc.) missing

Lines 103/104 – constatation “For zucchini plants, ToLCNDV-India infected plants showed heavier curling and mosaic symptoms than those of ToLCNDV-ES infected plants, demonstrating that ToLCNDV-ES had lower pathogenicity than ToLCNDV-India in this crop” is not evident from the descriptions provided in the Table 1 where all symptoms descriptions for zucchini are the same.

Figure 4. – The photos provided of the infected plants are too small and on some it appears that the symptoms are present on the negative controls but not on the infected plants.

Table 1. consider writing the cultivars like this: Pumpkin (cv. Shin Aisa Matdol);

line 115 – infected plants/ number of inoculated plants

Line 122 – in cucurbit crops.

Figure 5 – watermelon cultivars inoculated with two ToLCNDV isolates

Table 2. No need for Symptom column – can be deleted

Line 130 – see comment for line 115

Discussion

Line 136 – in 2012 in Spain – please add the reference under the 5 and 11 as supporting material 

Line 143 – Agrobacterium (italic)

Line 153 – ToLCNDV induced disease.

Line 158 – delete , after pumpkin

Line 160 – delete pathogens

Line 161 – cultivars resistant or tolerant to ToLCNDV infection.

Materials and Methods

Line 164 – Virus sources

Line 168 – Korea). Rolling

Line 172 – NCBI can be omitted/deleted

Section 4.2 should be explained with more details concerning isolates used etc.

Line 181 – Table S2 does not exist

Line 191 – C degrees

Line 192 – was verified by conventional PCR

Line 196 – uniform names of primers used (not identical in main text and supplementary material); please indicate how the quality of isolated DNA was checked to avoid false negative PCR results due to poor DNA isolation/quality.

Line 207 – section Samples…. blotting (36) can be deleted, together with reference 36 (too wide/  nonspecific)

Reviewer 3 Report

The manuscript "Different infectivity of tomato leaf curl New Delhi virus the Mediterranean and Southern 2 Asian isolates in cucurbit crops" is a new step to an understanding of the pathogenicity of ToLCNDV in different species. The experiments and results are nicely presented. But, the no of plants used for inoculation is a little less. 

Figure 1 legend. Can the authors write here, how many bootstraps were used?

Line 82.  Authors should write exactly or approximately size of multimers of infectious clone for e.g. 1.2, 1.5 mer. Use ~ if want to write approximately based on the marker loaded in the gel.

Figure 3 and 4, panel B. The intensity of PCR band appears saturated due to more cycles of PCR and which apparently does not allow one to visualize the differences of virus titre across different samples. Mere detection is not the purpose here. I suggest authors to perform a low number of PCR cycles or run a q-PCR on these samples.

Table 1. Line 114. No of plant tested is less. At least 10-15 should be used per species.
